

# Supply-side barriers to maternal health care utilization at health sub-centers in India

Aditya Singh

Global Health and Social Care Unit, School of Health Sciences and Social Work, University of Portsmouth, Portsmouth, United Kingdom

## ABSTRACT

**Introduction**. There exist several barriers to maternal health service utilization in developing countries. Most of the previous studies conducted in India have focused on demand-side barriers, while only a few have touched upon supply-side barriers. None of the previous studies in India have investigated the factors that affect maternal health care utilization at health sub-centers (HSCs) in India, despite the fact that these institutions, which are the geographically closest available public health care facilities in rural areas, play a significant role in providing affordable maternal health care. Therefore, this study aims to examine the supply-side determinants of maternal service utilization at HSCs in rural India.

**Data and Methods**. This study uses health facility data from the nationally representative District-Level Household Survey, which was administered in 2007–2008 to examine the effect of supply-side variables on the utilization of maternal health care services across HSCs in rural India. Since the dependent variables (the number of antenatal registrations, in-facility deliveries, and postnatal care services) are count variables and exhibit considerable variability, the data were analyzed using negative binomial regression instead of Poisson regression.

**Results**. The results show that those HSCs run by a contractual auxiliary nurse midwife (ANM) are likely to offer a lower volume of services when compared to those run by a permanent ANM. The availability of obstetric drugs, weighing scales, and blood pressure equipment is associated with the increased utilization of antenatal and postnatal services. The unavailability of a labor/examination table and bed screen is associated with a reduction in the number of deliveries and postnatal services. The utilization of services is expected to increase if essential facilities, such as water, telephones, toilets, and electricity, are available at the HSCs. Monitoring of ANM's work by Village Health and Sanitation Committee (VHSC) and providing in-service training to ANM appear to have positive impacts on service utilization. The distance of ANM's actual residence from the sub-center village where she works is negatively associated with the utilization of delivery and postnatal services. These findings are robust to the inclusion of several demand-side factors.

**Conclusion**. To improve maternal health care utilization at HSCs, the government should ensure the availability of basic infrastructure, drugs, and equipment at all locations. Monitoring of the ANMs' work by VHSCs could play an important role in improving health care utilization at the HSCs; therefore, it is important to establish VHSCs in each sub-center village. The relatively low utilization of maternity services in those HSCs that are run solely by contractual ANMs requires further investigation.

Corresponding author
Aditya Singh, aadigeog@gmail.com

## INTRODUCTION

Although India appears to have missed its Millennium Development Goal (MDG)-5 target, which aimed to reduce maternal mortality by two-thirds by the year 2015, the fact remains that it has witnessed a considerable decline in maternal mortality over last 15 years or so (*United Nations, 2016*). This decline is a result of concerted efforts made by the Government of India to improve financial and geographic access to maternal health care services through the publicly-funded health system, especially in rural areas (*Kumar, 2010*). Unfortunately, the utilization of maternal health care services is still low. For instance, the proportions of Indian women who received full antenatal care, who had a successful delivery, and who underwent a postnatal check-up were 18%, 51%, and 49%, respectively, in 2007–2008. The situation in rural areas is even worse (*International Institute for Population Sciences, 2010*).

Health service utilization depends on a number of factors. These factors can be understood through the demand–supply framework (*Ensor & Cooper, 2004*). In the demand–supply framework, demand-side determinants are defined as individual, household, or community characteristics that influence the demand for health services. These factors may operate at the individual, household, or community level. In contrast, supply-side factors are those characteristics of the health system that exist beyond the control of potential health service users, such as health facilities, drugs, equipment, finances, human resources, geographic distance, and so on (*Peters et al., 2008*).

A number of previous studies have identified several demand-side factors as important obstacles to health care utilization in developing countries (*Ensor & Cooper, 2004*; *Kesterton et al., 2010*; *O'Donnell, 2007*; *Sarma, 2009*). However, only a few have truly addressed supply-side barriers (*Metcalfe & Adegoke, 2013*). Although many studies have documented the shortages of drugs, equipment, physical infrastructure, finances, and human resources in the provision of health care (*Bajpai, Dholakia & Sachs, 2008*; *Bajpai, 2014*; *Bhandari & Dutta, 2007*; *Hazarika, 2013*; *Raut-Marathe, Sardeshpande & Yakkundi, 2015*; *Varatharajan, Thankappan & Jayapalan, 2004*), the evidence on how these supply-side factors affect service provision at publicly funded rural health facilities in India is still very sparse (*Kumar & Dansereau, 2014*).

A health sub-center (HSC) serves as the first point of contact between the government-funded health system and those individuals living in rural areas. HSCs play an important role in bringing about behavioral changes through interpersonal communication, and they provide a wide range of services, including maternal and child health (MCH) care, family planning, nutrition, and immunization services (*Bhandari & Dutta, 2007*). The auxiliary nurse midwife (ANM), the sole female functionary at the HSC level, is largely responsible for implementing maternal health programs in rural India (*Malik, 2009*; *Mavalankar & Vora, 2008*). Unfortunately, the level maternal service utilization at HSCs in most states is still very low (*International Institute for Population Sciences, 2010*). Therefore, it is imperative

to examine the factors responsible for the low utilization of maternal health care services at this delivery point. In this regard, supply-side factors (health workers, drugs, equipment, infrastructure, and training) are very important from a policy perspective, as managing these factors is far easier and more effective than manipulating the factors that are known to improve the demand for health services. Hence, using data from a large, nationally representative, cross-sectional household and health facility survey, this study aims to examine the effect of supply-side factors on the utilization of maternal health services at the HSC level.

## DATA AND METHODS

### What is a health sub-center?

The public health care system in rural areas is a three-tiered hierarchical system; HSCs are located at the bottom tier, primary health centers (PHCs) are located at the middle tier, and community health centers (CHCs) are featured at the top tier. CHCs are 30-bed rural hospitals that are equipped with specialists, doctors, nurses, technicians, and other facilities to provide effective referral support to a population of about 80,000–120,000 individuals. At the middle of the hierarchy is the PHC, which serves a population of about 30,000–50,000 people. It is usually staffed with a qualified allopathic doctor, a pharmacist, and a lab technician (*Government of India, 2007*).

HSCs are entrusted with a number of curative, preventive, and promotional health care-related tasks. They are expected to bring about behavioral changes through interpersonal communication, and they provide basic services related to MCH, family planning, nutrition, immunization, communicable diseases, and so on (*Government of India, 2007*). Each HSC is staffed with a female health worker, officially known as an ANM, and a male health worker (MHW), also known as a male multipurpose worker. In some HSCs, two ANMs are posted. As per Indian Public Health Standards (IPHS), each HSC should have at least four workers–two ANMs, one MHW, and a support worker (*Pallikadavath et al., 2013*).

### Data source

This study used data from the third round of the District Level Household and Facility Survey (DLHS-3), a cross-sectional survey that was administered in 2007–2008. The survey was designed to collect data on various aspects of MCH service utilization, and it aimed to assess the capacity and preparedness of publicly funded health facilities in terms of their infrastructure, human resources, drugs, equipment, and training. Data on the first aspect were collected through a household survey, while data on health facility capacity and preparedness were collected through a facility survey. Apart from collecting information on the resources available at these health facilities, the facility survey also gathered data on the utilization of MCH services at the HSCs during the month preceding the survey.

The DLHS-3 used a multistage, stratified probability proportional to size (PPS) sampling design. In each district, 50 primary sampling units (PSUs), or census villages, were selected during the first stage via systematic PPS sampling. From each PSU, households were selected using circular systematic sampling. The selection of rural health facilities for the Facility Survey was linked with the sampled rural PSUs. The Facility Survey included only those
PHCs and HSCs that were located in proximity to, and which were expected to serve, the health care needs of the sampled rural PSUs. All CHCs and District Hospitals (DHs) were included in the survey (*International Institute for Population Sciences, 2010*). The survey collected data via face-to-face interviews. In the Facility Survey, some information was collected directly from the official registers, which were maintained by the health facility itself. In all, the DHLS-3 household survey collected information from 720,320 households; there were 643,944 ever-married women aged 15–49 years and 166,260 unmarried women aged 15–24 years. Conversely, the Facility Survey from the DLHS-3 surveyed 18,068 HSCs, 8,619 PHCs, 4,162 CHCs, and 594 district hospitals. The overall household response rate was around 94%. The overall response rates for ever-married women and unmarried women were 89% and 85%, respectively. The survey oversampled households by 10%, which would serve as a cushion to protect against the effects of non-responders. More information on the sampling procedures, survey design implementation, and response rates for the DHLS-3 are found in the DLHS-3 national report, which can be found at http://rchiips.org/pdf/INDIA_REPORT_DLHS-3.pdf.

This study used information on 18,068 HSCs to examine the association between a number of supply-side factors (health personnel, infrastructure, medical equipment, drug availability, and so on) and the delivery of maternal health care services. As per the DLHS-3 report, about 91% of HSCs had a permanent ANM in place, while about 20% had a contractual ANM. The average population served by the sampled HSCs was 8,372 individuals (*International Institute for Population Sciences, 2010*).

## Dependent variables

This study examined MCH service utilization at HSCs using three dependent variables– namely, the number of antenatal care registrations (a proxy for antenatal care utilization), the number of deliveries conducted, and the number of postnatal care services provided in each facility during the month preceding the survey. Thus, the exposure period was the same for all health facilities. All of the aforementioned variables were count variables. The proportion of HSCs with zero antenatal care registrations and postnatal care services was about 3% and 9%, respectively. About 76% of facilities did not perform any deliveries in the month preceding the survey. The average numbers of women who received antenatal care, underwent deliveries, and provided with postnatal care were 15, one, and seven, respectively.

## Independent variables

The selection of independent variables in this study was guided by a demand–supply framework provided by *Ensor & Cooper (2004)*. The study included a number of supply- and demand-side variables that were available in the DLHS-3 dataset (see Tables 1A and 1B). Most of these variables were found to be associated with health care utilization in various settings around the world. For instance, the presence of an ANM was found to be associated with an increase in maternal health care utilization by a study conducted in South India (*Navaneetham & Dharmalingam, 2002*). Similarly, other supply- side factors such as drugs, physical infrastructure, in-service training, equipment, and

**Table 1A** Distribution of health sub-center characteristics, 2007–2008 ($n = 17,667$).

| Independent variables | | n | Percent |
|---|---|---|---|
| **Health personnel** | Auxiliary Nurse Midwife (ANM) | | |
| | None | 624 | 3.53 |
| | Only Permanent ANM | 13,486 | 76.33 |
| | Only Contractual ANM | 756 | 4.28 |
| | Both | 2,801 | 15.85 |
| **Drug availability** | Paracetamol | | |
| | Yes | 12,576 | 71.43 |
| | No | 5,030 | 28.57 |
| | Iron and folic acid | | |
| | Yes | 12,825 | 72.72 |
| | No | 4,811 | 27.28 |
| **Equipment** | Blood pressure instrument | | |
| | Yes | 13,884 | 78.59 |
| | No | 3,783 | 21.41 |
| | Weighing scale | | |
| | Yes | 14,101 | 79.82 |
| | No | 3,566 | 20.18 |
| | Sims speculum | | |
| | Yes | 10,666 | 60.37 |
| | No | 7,001 | 39.63 |
| | Examination table | | |
| | Available and usable | 9,598 | 54.33 |
| | Available and unusable | 996 | 5.64 |
| | Not available | 7,073 | 40.04 |
| | Labor table | | |
| | Available and usable | 5,420 | 30.68 |
| | Available and unusable | 1,028 | 5.82 |
| | Not available | 11,219 | 63.5 |
| | Bed screen | | |
| | Available and usable | 4,032 | 22.82 |
| | Available and unusable | 643 | 3.64 |
| | Not available | 12,992 | 73.54 |
| **Infrastructure** | Electricity | | |
| | Regular supply | 3,756 | 21.26 |
| | Irregular supply | 5,676 | 32.13 |
| | No connection | 8,235 | 46.61 |
| | Water supply | | |
| | No | 4,721 | 26.72 |
| | Yes | 12,946 | 73.28 |
| | Toilet | | |
| | No | 9,105 | 51.54 |
| | Yes | 8,562 | 48.46 |
| | Telephone | | |
| | No | 15,979 | 90.45 |
| | Yes | 1,688 | 9.55 |

**Table 1A** (*continued*)

| Independent variables | | n | Percent |
|---|---|---|---|
| **Quality variables** | Integrated skills development training | | |
| | No | 9,034 | 51.13 |
| | Yes | 8,633 | 48.87 |
| | Skilled birth training | | |
| | No | 11,886 | 67.28 |
| | Yes | 5,781 | 32.72 |
| | Monitoring by VHSC | | |
| | No | 7,000 | 39.62 |
| | Yes | 10,667 | 60.38 |
| **Other variables** | Region | | |
| | North | 1,911 | 10.82 |
| | Central | 7,511 | 42.51 |
| | Northeast | 1,451 | 8.21 |
| | East | 1,815 | 10.27 |
| | West | 1,845 | 10.44 |
| | South | 3,134 | 17.74 |
| | ANM's residence from SC (km) | | |
| | <4 | 7,197 | 41.07 |
| | 5–20 | 6,923 | 39.51 |
| | 21–40 | 2,814 | 16.06 |
| | >40 | 588 | 3.36 |

**Table 1B** Descriptive statistics of the variables ($n = 17,667$).

| Variable | Mean/Percent | SD | Minimum | Maximum |
|---|---|---|---|---|
| Essential obstetric drug | 1.5 | 1.7 | 0 | 5 |
| Log of catchment population | 8.6 | 0.7 | 1.6 | 13.8 |
| % population in lowest wealth quintile | 3.4 | 6.2 | 0.0 | 49.5 |
| Total fertility rate | 3.1 | 1.0 | 1.4 | 5.9 |
| % Hindu population | 81.6 | 24.3 | 0.0 | 100.0 |
| Average years of schooling among mothers | 4.4 | 2.4 | 0.5 | 11.5 |

accessibility were also found to be influential in shaping health care utilization patterns (*Agrawal et al., 2012*; *Blankart, 2012*; *Kesterton et al., 2010*; *Kumar & Dansereau, 2014*; *Kumar & Prakash, 2013*; *Peters et al., 2008*; *Shaikh & Hatcher, 2005*; *Valdivia, 2002*; *Yeager, 2012*). These variables can be divided into seven categories: (a) health personnel, (b) drugs, (c) equipment, (d) infrastructure, (e) quality-related variables, (f) geographic variables, and (g) socioeconomic and demographic variables. Detailed descriptions of the variables included in the analysis are given in the following paragraphs.

An HSC is ideally supposed to have two ANMs (one permanent and another contractual) and an MHW. The DLHS-3 featured a question that asked whether a position for a specific type of worker at a given HSC was filled. Using this information (which involved combining permanent and contractual ANM variables), a health worker variable with four

categories–namely, HSC without an ANM, HSC with only a permanent ANM (reference category), HSC with only a contractual ANM, and HSC with both ANMs–was generated. The availability of drugs was captured using two binary variables–one for iron–folic acid (IFA) tablets/syrup and another for paracetamol tablets. An index was created to capture the availability of essential obstetric care drugs. The index featured scores that ranged from 0 to 5, with HSCs receiving one point for having each of the following obstetric care drugs: gentamycin, magnesium sulfate, ampicillin, metronidazole, and misoprostol. To measure the availability of medical equipment, the following binary variables were used: blood pressure instrument, weighing scale for adults, and Sims speculum. Three other categorical variables–the availability of a labor table, an examination table, and a bed screen (using the following categories: available and functional (reference category), available but unusable, and not available)–were also included in the analysis.

The following variables were included to capture the availability of basic amenities and infrastructure: a categorical variable for the supply of electricity (using these categories: 'regular supply' (reference category), 'irregular supply,' and 'no electricity connection'), as well as three binary variables (water, toilets, and telephone facilities). To capture the quality of a given health facility, and to check whether high-quality facilities were likely to deliver a higher volume of services, three additional binary variables were included. The first two of these binary variables captured the quality of an ANM by asking whether the ANM received in-service 'integrated skills development training' and 'skilled birth attendance training' at any time during the last five years preceding the survey. The third binary variable that captured the quality of HSCs was whether the VHSC monitored the ANMs' work on a regular basis.

The main purpose of this study was to examine the effect of supply-side factors on the delivery of maternal health care services at the HSC level. However, it also included a number of demand-side variables that were previously found to affect the demand for health care services, such as socioeconomic, demographic, and geographical factors (*Metcalfe & Adegoke, 2013*; *Midhet, Becker & Berendes, 1998*; *Patel & Ladusingh, 2015*; *Singh, Kumar & Pranjali, 2014*; *Singh et al., 2012*; *Sunil, Rajaram & Zottarelli, 2006*; *Ten Hoope-Bender, Liljestrand & MacDonagh, 2006*). All of these variables were calculated for the district as a whole. These variables included a log of the population living in the facility's catchment area, a district-level fertility indicator (the total fertility rate was calculated using birth-order information obtained from the DLHS-3 ever-married women dataset), the average number of years of education among mothers, the percentage of households in the lowest economic quartile (as defined by an asset score calculated from the DLHS-3 household file), and the percentage of Hindu households in a given district. All of the aforementioned variables were continuous variables. The analysis also included a categorical variable that captured the effect of the distance of the ANMs' actual residence from the sub-center village where they work. The variable has four categories: 'within 4 kilometers' (reference category), '5–20 kilometers,' '21–40 kilometers', and 'more than 40 kilometers.'

In this study, a 'region' represents a group of Indian states. The north region (1), which served as the reference category for the 'region' variable, includes Jammu and Kashmir, Himachal Pradesh, Punjab, Rajasthan, Haryana, Chandigarh (Union Territory –UT), and

Delhi; the central region (2) includes the states of Uttar Pradesh, Uttaranchal, Madhya Pradesh, and Chhattisgarh; the northeast region (3) includes the states of Sikkim, Assam, Meghalaya, Manipur, Mizoram, Nagaland, Tripura, and Arunachal Pradesh; the east region (4) includes the states of Bihar, Jharkhand, West Bengal, and Orissa; the west region (5) includes the states of Gujarat, Maharashtra, Goa, and the UTs of Dadra and Nagar Haveli, and Daman and Diu; and the south region (6) includes the states of Kerala, Karnataka, Andhra Pradesh, Tamil Nadu, and the UTs of Andaman and Nicobar Islands, Pondicherry, and Lakshadweep.

## Statistical analysis

The study used negative binomial regression to examine the association between the availability of various resources at the HSCs and the volume of the maternal health care services provided during the last month preceding the survey. Since the dependent variables consisted of non-negative count data, a negative binomial regression technique was chosen for the analysis (*Coxe, West & Aiken, 2009*). The possibility of using a Poisson regression model was ruled out due to its strict assumption that the conditional mean and variance of the dependent variable should be the same (*Land, McCall & Nagin, 1996*). None of the dependent variables fulfilled this requirement; all of them exhibited clear over-dispersion. For instance, the 'antenatal registration' variable had a mean of 14.7 and a variance of 364.5. Similarly, delivery (with a mean of 1.1 and a variance of 10) and postnatal care (with a mean of 7.3 and a variance of 56.4) also exhibited over-dispersion. In such cases, a Poisson regression model would usually produce inefficient estimates. A negative binomial model does not require equality between the mean and variance; thus, it also enabled us to take into account unmeasured characteristics that generated over-dispersion in the count data (*Gardner, Mulvey & Shaw, 1995*). Hence, a negative binomial model was preferred over a Poisson model for the current analysis.

The generalized form of the negative binomial regression model used in this study can be written as follows:

$$Y_{fd} = \beta_0 + \beta_1 X_1 + \beta_2 X_2 + \beta_3 X_3 + \ldots + \beta_k X_k + (\sigma_d) + \varepsilon_{fd},$$

Where, $Y$ is the outcome variable (the number of health services delivered during the last month preceding the survey) at sub-center $f$ in district $d$, $X_n$ represents the facility and district-level demographic factors affecting the outcome, and $\varepsilon_{fd}$ is the error term and $\sigma_d$ is the random effect that varies at the district level. The random intercept $\sigma_d$ captures the effect of unobserved district-specific characteristics that cause some districts to have facilities that produce a greater volume of services than others, such as the presence of fewer private health facilities or the socioeconomic and demographic characteristics of a given population; these factors can potentially affect the demand for maternity services. The catchment population of HSC served as an 'exposure variable' in all models. Initially, individual covariates were tested separately for their association with the dependent variables; the results are given under the 'unadjusted' columns in Tables 2–4. Following that, full regression models featuring all covariates were run to obtain 'adjusted' coefficients.

**Table 2** Unadjusted and adjusted incidence rate ratios for the volume of antenatal care registrations at health sub-centers in India, 2007–2008 ($n = 16,537$).

| | Outcome: number of antenatal registrations. | | | | | |
|---|---|---|---|---|---|---|
| | **Unadjusted** | | | **Adjusted** | | |
| | IRR | *P*-value | 95% CI | IRR | *P*-value | 95% CI |
| **Health personnel** | | | | | | |
| **Auxiliary Nurse Midwife (ANM)** | | | | | | |
| None | 0.795 | 0.000 | 0.745, 0.849 | 0.846 | 0.001 | 0.763, 0.938 |
| Only Contractual ANM | 0.923 | 0.021 | 0.862, 0.988 | 0.901 | 0.014 | 0.829, 0.979 |
| Both | 1.085 | 0.000 | 1.043, 1.128 | 1.042 | 0.178 | 0.982, 1.105 |
| **Drug availability** | | | | | | |
| **Paracetamol** | | | | | | |
| No | 1.002 | 0.884 | 0.972, 1.034 | 0.997 | 0.879 | 0.954, 1.041 |
| **Iron and folic acid** | | | | | | |
| No | 1.013 | 0.464 | 0.979, 1.048 | 0.992 | 0.735 | 0.946, 1.040 |
| **Equipment** | | | | | | |
| **BP instrument** | | | | | | |
| Yes | 1.045 | 0.002 | 1.016, 1.075 | 1.047 | 0.029 | 1.005, 1.092 |
| **Weighing scale** | | | | | | |
| Yes | 1.044 | 0.004 | 1.014, 1.074 | 1.023 | 0.276 | 0.982, 1.066 |
| **Examination table** | | | | | | |
| Available but unusable | 0.926 | 0.003 | 0.880, 0.975 | 0.943 | 0.130 | 0.874, 1.017 |
| Not available | 0.921 | 0.000 | 0.896, 0.947 | 0.956 | 0.025 | 0.919, 0.994 |
| **Bed screen** | | | | | | |
| Available but unusable | 0.940 | 0.064 | 0.881, 1.003 | 0.968 | 0.477 | 0.886, 1.058 |
| Not available | 0.930 | 0.000 | 0.903, 0.958 | 0.956 | 0.032 | 0.917, 0.996 |
| **Infrastructure** | | | | | | |
| **Electricity** | | | | | | |
| Irregular supply | 0.926 | 0.000 | 0.891, 0.963 | 0.933 | 0.011 | 0.885, 0.984 |
| No connection | 0.896 | 0.000 | 0.864, 0.929 | 0.915 | 0.000 | 0.871, 0.962 |
| **Water supply** | | | | | | |
| Yes | 1.056 | 0.000 | 1.028, 1.085 | 1.026 | 0.180 | 0.988, 1.066 |
| **Toilet** | | | | | | |
| Yes | 1.045 | 0.001 | 1.019, 1.072 | 0.997 | 0.882 | 0.961, 1.034 |
| **Telephone** | | | | | | |
| Yes | 1.050 | 0.164 | 0.980, 1.125 | 1.039 | 0.443 | 0.942, 1.145 |
| **Quality variables** | | | | | | |
| **ISD training in last five years** | | | | | | |
| Yes | 1.036 | 0.004 | 1.011, 1.062 | 1.014 | 0.416 | 0.980, 1.050 |
| **VHSC monitoring work** | | | | | | |
| Yes | 1.037 | 0.010 | 1.009, 1.066 | 1.054 | 0.010 | 1.013, 1.096 |

**Table 2** (*continued*)

| | Outcome: number of antenatal registrations. | | | | | |
|---|---|---|---|---|---|---|
| | **Unadjusted** | | | **Adjusted** | | |
| | IRR | *P*-value | 95% CI | IRR | *P*-value | 95% CI |
| **Other variables** | | | | | | |
| **Region** | | | | | | |
| Central | 1.458 | 0.000 | 1.280, 1.661 | 0.978 | 0.792 | 0.832, 1.151 |
| Northeast | 0.943 | 0.476 | 0.801, 1.109 | 1.015 | 0.845 | 0.871, 1.184 |
| East | 1.576 | 0.000 | 1.328, 1.870 | 1.006 | 0.941 | 0.849, 1.193 |
| West | 1.232 | 0.013 | 1.044, 1.455 | 0.993 | 0.924 | 0.860, 1.146 |
| South | 0.947 | 0.479 | 0.815, 1.101 | 0.801 | 0.004 | 0.689, 0.931 |
| **ANM's residence from SC (in km)** | | | | | | |
| 5–20 | 0.974 | 0.172 | 0.938, 1.011 | 0.994 | 0.765 | 0.956, 1.033 |
| 21–40 | 0.916 | 0.011 | 0.857, 0.980 | 0.940 | 0.071 | 0.879, 1.005 |
| >40 | 0.882 | 0.063 | 0.772, 1.007 | 0.906 | 0.152 | 0.792, 1.037 |
| **Log of catchment population** | 1.303 | 0.000 | 1.275, 1.332 | 1.326 | 0.000 | 1.267, 1.386 |
| **Socioeconomic variables** | | | | | | |
| **% population in lowest wealth quintile** | 0.993 | 0.051 | 0.986, 1.000 | 1.004 | 0.149 | 0.999, 1.009 |
| **Total fertility rate** | 1.230 | 0.000 | 1.183, 1.280 | 1.112 | 0.000 | 1.051, 1.176 |
| **% Hindu population** | 1.006 | 0.000 | 1.004, 1.007 | 1.004 | 0.000 | 1.002, 1.005 |
| **Maternal education (in years)** | 0.910 | 0.000 | 0.895, 0.925 | 0.945 | 0.000 | 0.926, 0.964 |
| **Dispersion parameter (alpha)** | | | | 0.444 | 0.000 | 0.417, 0.472 |
| **Likelihood-ratio test of alpha = 0: Chi$^2$Statistic = 2230.01 ($p = 0.000$)** | | | | | | |

**Notes.**

IRR, Incidence Rate Ratio; CI, Confidence Interval; ANM, Auxiliary Nurse Midwife; VHSC, Village Health and Sanitation Committee; SC, Sub-Centre/Health Sub-Centre; BP, Blood Pressure; ISDT, Integrated Skills Development Training.

Variance Inflation Factor –1.61 (for detailed VIF, see Table S1).

After the regression, the residuals from each model were tested for the presence of heteroscedasticity using Park's test. The test results of all three models confirmed the presence of heteroscedasticity (results not shown; available on demand). In the presence of heteroscedasticity, although the estimated coefficient remained unbiased and consistent, the estimated standard errors were not reliable. Hence, White's robust procedure (using the 'robust' command in Stata) was applied to obtain robust standard errors.

Alpha, the over-dispersion parameter, is presented at the bottom of each model in Tables 2–4. If alpha is zero, then the data are not over-dispersed and a Poisson model is suitable. If alpha is greater than zero, then the data are over-dispersed and the negative binomial distribution is able to model the data more accurately than Poisson distribution. In order to test that the dispersion parameter, alpha, is equal to zero, a likelihood ratio chi-square test was applied; the results are also presented in Tables 2–4. The presence of large test statistics with very small P-values suggests that the response variables were over-dispersed and that they were better estimated using negative binomial regression than a Poisson model.

All models were tested for any potential multicollinearity among the independent variables; the variance inflation factor (VIF) was used as a post-estimation procedure. The overall VIFs for the models were very small (see the footnotes in Tables 2–4),

**Table 3 Unadjusted and adjusted incidence rate ratios for delivery services at health sub-centers in India, 2007–2008. ($n = 16,030$).**

| | Outcome: number of deliveries conducted | | | | | |
|---|---|---|---|---|---|---|
| | **Unadjusted** | | | **Adjusted** | | |
| | IRR | P-value | 95% CI | IRR | P-value | 95% CI |
| **Health personnel** | | | | | | |
| **Auxiliary Nurse Midwife (ANM)** | | | | | | |
| None | 0.496 | 0.000 | 0.376, 0.654 | 0.579 | 0.000 | 0.441, 0.760 |
| Only Contractual ANM | 0.640 | 0.002 | 0.484, 0.846 | 0.709 | 0.017 | 0.534, 0.940 |
| Both | 0.874 | 0.087 | 0.749, 1.020 | 0.863 | 0.053 | 0.743, 1.002 |
| **Drug availability** | | | | | | |
| **Essential obstetric drugs** | 1.103 | 0.000 | 1.067, 1.141 | 1.060 | 0.000 | 1.026, 1.096 |
| **Equipment** | | | | | | |
| **Sims speculum** | | | | | | |
| No | 0.685 | 0.000 | 0.616, 0.762 | 0.901 | 0.053 | 0.811, 1.002 |
| **Labor table** | | | | | | |
| Available but unusable | 0.569 | 0.000 | 0.475, 0.682 | 0.652 | 0.000 | 0.545, 0.780 |
| Not available | 0.314 | 0.000 | 0.284, 0.348 | 0.438 | 0.000 | 0.394, 0.487 |
| **Bed Screen** | | | | | | |
| Available but unusable | 0.805 | 0.072 | 0.636, 1.019 | 0.970 | 0.793 | 0.771, 1.220 |
| Not available | 0.474 | 0.000 | 0.425, 0.528 | 0.694 | 0.000 | 0.623, 0.774 |
| **Infrastructure** | | | | | | |
| **Electricity** | | | | | | |
| Irregular supply | 0.879 | 0.077 | 0.761, 1.014 | 0.934 | 0.342 | 0.813, 1.075 |
| No connection | 0.470 | 0.000 | 0.410, 0.539 | 0.678 | 0.000 | 0.590, 0.779 |
| **Water supply** | | | | | | |
| Yes | 1.726 | 0.000 | 1.546, 1.928 | 1.200 | 0.001 | 1.073, 1.342 |
| **Toilet** | | | | | | |
| Yes | 2.100 | 0.000 | 1.912, 2.306 | 1.440 | 0.000 | 1.306, 1.588 |
| **Telephone** | | | | | | |
| Yes | 1.695 | 0.000 | 1.354, 2.122 | 1.307 | 0.015 | 1.054, 1.621 |
| **Quality variables** | | | | | | |
| **SBA training in last five years** | | | | | | |
| Yes | 1.486 | 0.000 | 1.350, 1.637 | 1.285 | 0.000 | 1.168, 1.413 |
| **VHSC monitoring work** | | | | | | |
| Yes | 1.086 | 0.121 | 0.978, 1.206 | 0.942 | 0.250 | 0.851, 1.043 |
| **Other variables** | | | | | | |
| **Region** | | | | | | |
| Central | 2.517 | 0.000 | 1.692, 3.744 | 0.914 | 0.695 | 0.581, 1.435 |
| Northeast | 0.685 | 0.146 | 0.411, 1.141 | 0.599 | 0.032 | 0.375, 0.956 |
| East | 0.338 | 0.000 | 0.196, 0.583 | 0.310 | 0.000 | 0.177, 0.541 |
| West | 2.680 | 0.000 | 1.631, 4.402 | 1.374 | 0.188 | 0.856, 2.206 |
| South | 0.828 | 0.421 | 0.522, 1.312 | 0.578 | 0.018 | 0.367, 0.910 |

**Table 3** (*continued*)

| | Outcome: number of deliveries conducted | | | | | |
|---|---|---|---|---|---|---|
| | Unadjusted | | | Adjusted | | |
| | IRR | *P*-value | 95% CI | IRR | *P*-value | 95% CI |
| ANM's residence from SC (in km) | | | | | | |
|    5–20 | 0.480 | 0.000 | 0.432, 0.533 | 0.643 | 0.000 | 0.580, 0.713 |
|    21–40 | 0.473 | 0.000 | 0.388, 0.577 | 0.637 | 0.000 | 0.525, 0.773 |
|    >40 | 0.482 | 0.000 | 0.327, 0.709 | 0.707 | 0.067 | 0.488, 1.025 |
| Log of catchment population | 1.198 | 0.000 | 1.097, 1.307 | 1.111 | 0.016 | 1.020, 1.210 |
| Socioeconomic variables | | | | | | |
| % population in lowest wealth quintile | 0.995 | 0.677 | 0.974, 1.017 | 1.011 | 0.265 | 0.992, 1.030 |
| Total fertility rate | 1.292 | 0.000 | 1.133, 1.473 | 1.144 | 0.095 | 0.977, 1.339 |
| % Hindu population | 1.017 | 0.000 | 1.012, 1.022 | 1.013 | 0.000 | 1.007, 1.018 |
| Maternal education (in years) | 0.861 | 0.000 | 0.813, 0.911 | 0.846 | 0.000 | 0.788, 0.908 |
| Dispersion parameter (alpha) | | | | 3.959 | 0.000 | 3.765, 4.162 |
| Likelihood-ratio test of alpha = 0: Chi$^2$ Statistic = 990.07 ($p$ = 0.000) | | | | | | |

**Notes.**

IRR, Incidence Rate Ratio; CI, Confidence Interval; ANM, Auxiliary Nurse Midwife; VHSC, Village Health and Sanitation Committee; SC, Sub-Centre/Health Sub-Centre; SBA, Skilled Birth Attendance.

Variance Inflation Factor −1.63 (for detailed VIF, see Table S2).

and wherever there was any evidence of high multicollinearity, the variables/categories responsible for causing this multicollinearity were dropped from the analysis. Tables S1–S3 present detailed VIFs for each model. The analysis was conducted using the Stata (version 12.0) statistical software (Stata Statistical Software, Release 12; StataCorp., College Station, TX, USA).

# RESULTS

In this study, the analytical samples did not include any missing or 'don't know' responses. Also, health facilities with missing, zero, and 'don't know' responses were not included in the analytical sample. As a result, the analytical sample was reduced to 17,537 HSCs for antenatal care, 16,030 HSCs for delivery care, and 17,112 HSCs for postnatal care.

The summary statistics for the dependent and independent variables are given in Tables 1A and 1B. Tables 2–4 report the results from the negative binomial regression models for antenatal registrations, deliveries, and postnatal services, respectively. The results are reported in the form of unadjusted and adjusted incidence rate ratios (IRRs), along with their 95% confidence intervals (CIs). An IRR greater than 1 implies that an increase in the dependent variable is associated with an increase in the outcome variable, and vice versa. The independent variables used in this study have been grouped under six previously described categories: health personnel, drug availability, equipment, infrastructure, facility quality, and the population's socioeconomic and demographic characteristics.

The results of the analysis revealed that those HSCs run solely by a contractual ANM were associated with a lower volume of maternal health care services when compared to those run solely by a permanent ANM. The delivery of antenatal and postnatal services in

**Table 4** Unadjusted and adjusted incidence rate ratios for postnatal care utilization at health sub-centers in India, 2007–08. ($n = 17,112$).

| | Outcome: number of postnatal care services provided | | | | | |
|---|---|---|---|---|---|---|
| | Unadjusted | | | Adjusted | | |
| | IRR | *P*-value | 95% CI | IRR | *P*-value | 95% CI |
| **Health personnel** | | | | | | |
| **Auxiliary Nurse Midwife (ANM)** | | | | | | |
| None | 0.783 | 0.000 | 0.698, 0.877 | 0.832 | 0.000 | 0.775, 0.894 |
| Only Contractual ANM | 0.887 | 0.007 | 0.813, 0.968 | 0.901 | 0.004 | 0.839, 0.968 |
| Both | 1.081 | 0.003 | 1.026, 1.138 | 1.052 | 0.014 | 1.010, 1.095 |
| **Drug availability** | | | | | | |
| **Essential obstetric drugs** | 1.002 | 0.753 | 0.989, 1.015 | 0.999 | 0.759 | 0.989, 1.008 |
| **Equipment** | | | | | | |
| **BP instrument** | | | | | | |
| Yes | 1.076 | 0.000 | 1.045, 1.109 | 1.053 | 0.001 | 1.022, 1.086 |
| **Weighing scale** | | | | | | |
| Yes | 1.068 | 0.000 | 1.036, 1.101 | 1.034 | 0.034 | 1.003, 1.067 |
| **Examination table** | | | | | | |
| Available but unusable | 0.935 | 0.014 | 0.887, 0.987 | 0.984 | 0.109 | 0.908, 1.010 |
| Not available | 0.911 | 0.000 | 0.886, 0.937 | 0.939 | 0.000 | 0.921, 0.976 |
| **Bed Screen** | | | | | | |
| Available but unusable | 0.951 | 0.142 | 0.889, 1.017 | 0.957 | 0.642 | 0.920, 1.052 |
| Not available | 0.904 | 0.000 | 0.876, 0.932 | 0.948 | 0.000 | 0.910, 0.969 |
| **Infrastructure** | | | | | | |
| **Electricity** | | | | | | |
| Irregular supply | 0.997 | 0.875 | 0.958, 1.038 | 1.000 | 0.995 | 0.961, 1.041 |
| No connection | 0.936 | 0.000 | 0.902, 0.971 | 0.973 | 0.162 | 0.936, 1.011 |
| **Water supply** | | | | | | |
| Yes | 1.024 | 0.099 | 0.996, 1.054 | 0.984 | 0.272 | 0.955, 1.013 |
| **Toilet** | | | | | | |
| Yes | 1.077 | 0.000 | 1.049, 1.105 | 1.029 | 0.040 | 1.001, 1.056 |
| **Telephone** | | | | | | |
| Yes | 1.163 | 0.000 | 1.083, 1.248 | 1.084 | 0.021 | 1.012, 1.162 |
| **Quality variables** | | | | | | |
| **ISD training in last five years** | | | | | | |
| Yes | 1.067 | 0.000 | 1.040, 1.094 | 1.030 | 0.028 | 1.003, 1.057 |
| **SBA training in last five years** | | | | | | |
| Yes | 1.075 | 0.000 | 1.046, 1.105 | 1.034 | 0.019 | 1.006, 1.064 |
| **VHSC monitoring work** | | | | | | |
| Yes | 1.092 | 0.000 | 1.061, 1.124 | 1.083 | 0.000 | 1.052, 1.114 |

**Table 4** (*continued*)

| | Outcome: number of postnatal care services provided | | | | | |
|---|---|---|---|---|---|---|
| | **Unadjusted** | | | **Adjusted** | | |
| | **IRR** | **P-value** | **95% CI** | **IRR** | **P-value** | **95% CI** |
| **Other variables** | | | | | | |
| **Region** | | | | | | |
| Central | 1.517 | 0.000 | 1.330, 1.732 | 1.149 | 0.063 | 0.992, 1.330 |
| Northeast | 0.893 | 0.182 | 0.757, 1.054 | 1.029 | 0.705 | 0.887, 1.194 |
| East | 1.599 | 0.000 | 1.345, 1.901 | 1.222 | 0.025 | 1.026, 1.455 |
| West | 1.561 | 0.000 | 1.320, 1.845 | 1.268 | 0.003 | 1.084, 1.482 |
| South | 1.458 | 0.000 | 1.253, 1.698 | 1.191 | 0.017 | 1.031, 1.376 |
| **ANM's residence from SC (in km)** | | | | | | |
| 5–20 | 0.941 | 0.000 | 0.914, 0.968 | 0.962 | 0.009 | 0.935, 0.991 |
| 21–40 | 0.901 | 0.000 | 0.868, 0.935 | 0.940 | 0.001 | 0.907, 0.976 |
| >40 | 0.853 | 0.000 | 0.795, 0.915 | 0.918 | 0.015 | 0.856, 0.984 |
| **Log of catchment population** | 1.350 | 0.000 | 1.317, 1.384 | 1.318 | 0.000 | 1.286, 1.352 |
| **Socioeconomic variables** | | | | | | |
| **% population in lowest wealth quintile** | 1.005 | 0.195 | 0.998, 1.012 | 1.007 | 0.024 | 1.001, 1.013 |
| **Total fertility rate** | 1.071 | 0.002 | 1.026, 1.117 | 1.011 | 0.675 | 0.960, 1.065 |
| **% Hindu population** | 1.008 | 0.000 | 1.006, 1.009 | 1.004 | 0.000 | 1.002, 1.006 |
| **Maternal education (in years)** | 0.949 | 0.000 | 0.933, 0.966 | 0.952 | 0.000 | 0.931, 0.973 |
| **Dispersion parameter (alpha)** | | | | 0.419 | 0.000 | 0.407, 0.432 |
| **Likelihood-ratio test of alpha = 0: Chi² Statistic = 1894.2 ( p = 0.000)** | | | | | | |

**Notes.**

IRR, Incidence Rate Ratio; CI, Confidence Interval; ANM, Auxiliary Nurse Midwife; VHSC, Village Health and Sanitation Committee; SC, Sub-Centre/Health Sub-Centre; ISDT, Integrated Skills Development Training; SBA, Skilled Birth Attendance.

Mean Variance Inflation Factor −1.63 (for detailed VIF, see Table S3).

those sub-centers that were solely run by a contractual ANM was about 8%–9% less than at those sub-centers run solely by a permanent ANM. The volume of deliveries in these HSCs decreased by about 30% [IRR = 0.709, $P = 0.017$]. The deployment of two ANMs was associated with a slight increase in the volume of postnatal services [IRR = 1.052, $P = 0.014$]. The HSCs without an ANM were expected to have a lower volume of antenatal registrations [IRR = 0.846, $P = 0.001$], deliveries [IRR = 0.579, $P = 0.000$], and postnatal services [IRR = 0.828, $P = 0.001$].

The availability of essential obstetric drugs was associated with increases in the volume of delivery services provided [IRR = 1.060, $P = 0.000$]. In the case of antenatal registrations, the availability of IFA tablets/syrup and paracetamol did not turn out to be statistically significant. The results also revealed that the availability of a blood pressure instrument was expected to slightly increase the volume of antenatal registrations [IRR = 1.047, $P = 0.029$] and postnatal care services [IRR = 1.053, $P = 0.001$]. The effect of the availability of an examination table on the volume of health service delivery was also statistically significant. In those HSCs where an examination table was not available, the volume of antenatal registrations and postnatal services decreased by about 4%–6%. A similar finding emerged in the case of labor table availability. The volume of deliveries that were conducted

decreased significantly if a labor table was 'available but not functional' [IRR = 0.652, $P = 0.000$] or not available at all [IRR = 0.438, $P = 0.000$]. Another variable that turned out to be statistically significant in the analysis was the unavailability of a bed screen. The unavailability of a bed screen decreased the volume of antenatal registrations and postnatal services by about 4%–5%. The negative effect of the unavailability of a bed screen was much stronger for delivery services [IRR = 0.694, $P = 0.000$].

A number of variables under the infrastructure category were significantly associated with the volume of maternal health services provided at the HSCs. Not having an electricity connection was associated with an approximately 32% decrease in the volume of deliveries [IRR = 0.678, $P = 0.000$] and a nearly 9.5% decrease in antenatal registrations [IRR = 0.915, $P = 0.000$]. The availability of water, telephones, and toilet facilities was likely to increase the volume of deliveries at sub-centers by about 20%, 30%, and 44%, respectively. The volume of postnatal services was also expected to increase in those HSCs that were equipped with telephone and toilet facilities.

In sub-centers where VHSCs monitored the ANMs' work, the volume of antenatal registrations and postnatal services was expected to increase by about 5% and 8%, respectively. Although ISDT was not associated with the volume of antenatal registrations, it was significantly associated with a slight increase in the volume of postnatal services [IRR = 1.030, $P = 0.028$]. Similarly, skilled birth attendance training (SBAT) was also associated with an increase in the volume of postnatal services [IRR = 1.034, $P = 0.019$] and deliveries [IRR = 1.285, $P = 0.000$]. Maternal health service utilization at sub-centers was found to be inversely associated with the distance of an ANM's actual residence from the sub-center village where she works. This was particularly true for delivery and postnatal services. For instance, at HSCs where the ANM lived 5–20 kilometers away from the sub-center village, the utilization of postnatal services decreased by about 4%, and if the ANM lived more than 40 kilometers away, the utilization reduced by about 9%.

## DISCUSSION

Using the most recent data available in the public domain, this study explored the association between sub-center characteristics and the volume of maternal health services delivered by those HSCs. The results suggest that the availability of essential obstetric drugs, blood pressure instruments, labor/examination tables, bed screens, and basic amenities such as water, electricity, toilets, etc., are the main supply-side factors that shape maternal health care utilization at sub-centers in India. The monitoring of an ANM's work by the VHSC, the provision of an ANM's in-service training, and the distance of the ANM's residence from the sub-center village also play important roles in maternal health care utilization at the sub-centers.

One significant finding of this study is that those sub-centers that are run solely by a contractual ANM delivered fewer services than those sub-centers run by a permanent ANM, despite controlling for other relevant factors. This applies to all three maternal health services considered in this study. This finding is particularly important because one of the key strategies of the National Rural Health Mission, which is an immediate measure that has

been implemented to increase the availability of health workers in publicly funded rural public health facilities, is to deploy contractual ANMs instead of recruiting permanent ANMs (*Dhingra & Dutta, 2011*). It is argued that recruiting contractual ANMs is both convenient and economical for the government (*Prinja et al., 2014*). However, this study indicates that this measure, despite its benefits to the government, does not seem to result in any substantial improvement in the utilization of maternity services. The results of our study show that the utilization of maternal health care services at HSCs run solely by a contractual ANM are not at par with those solely run by permanent ANMs. The volume of service delivery does not rise substantially, even when both ANMs run a sub-center. In other words, deploying a second ANM to sub-centers on a contractual basis is not likely to improve maternal health service utilization.

There may be a number of factors that may lead to the anomalous performance of ANMs in these two streams. Previous studies have recorded how low motivation is an important factor that underlies the poor performance of contractual workers. Workers' low motivation is often the result of a number of factors such as job insecurity, low salary, fewer privileges, discrimination between regular and contractual staff by higher authorities, the lack of a career path, and so on (*Kumar et al., 2014a*; *Kumar et al., 2014b*; *Kumar et al., 2013*). With respect to workers' salary, there is a significant difference in the pay offered to contractual and regular employees. Contractual ANMs receive much less remuneration for the same amount of work and responsibilities as a permanent ANM. For instance, a contractual ANM in the state of Bihar earns a meager ₹11,500 per month when compared to a permanent ANM, who receives a salary of ₹25,000 per month (*Government of Bihar, 2011*; *Government of Bihar, 2014*). A similar difference is found in other states as well. Therefore, the government and policymakers should look into this issue and develop a solution to ensure that the investment they are making in deploying contractual ANMs is optimally utilized.

The provision of quality maternal care services not only depends on skilled health personnel, but it is also reliant on the availability of essential drugs and supplies (*Yeager, 2012*). In line with previous studies, this investigation finds that the availability of essential obstetric drugs is strongly associated with a higher volume of delivery and postnatal care services at the HSCs (*Mkoka et al., 2014*). However, regrettably, about 42% of sub-centers are still devoid of all the essential obstetric drugs recommended by the Indian Public Health Standards. A lack of essential obstetric drugs at the facility may have huge implications for maternal mortality at the facility level. A recent study has concluded that increased facility deliveries in India do not contribute to a reduction in maternal deaths because these facilities often have a weak drug and medical supply system, which leads to poor obstetric care quality (*Randive, 2016*). The lack of drugs has also been found to cause distrust between users and health care providers; it creates a difficult working environment and decreases health workers' morale as well (*Mkoka et al., 2014*). All of these factors, when combined, could result in the low utilization of health care services.

The study also finds that the non-availability of a labor/examination table and bed screen significantly reduced the volume of services delivered at the sub-centers. This finding is also in line with the findings of a previous study (*Kumar & Dansereau, 2014*). Like labor
or examination tables, bed screens are also essential items; women do not feel comfortable during medical examinations when there are no bed screens. It must be noted that the descriptive results indicate that about 60% of sub-centers do not have a labor table and about 74% do not have a bed screen; furthermore, about 40% do not have an examination table. It is clear that there is plenty of room for improvement in these areas. Therefore, efforts should be made to equip sub-centers with these essential items.

In this study, the VHSC's ability to monitor the sub-center had a statistically significant effect on the antenatal and postnatal care services provided. This finding was consistent with those of a recent study, which concluded that the percentage of women seeking antenatal and postnatal care is higher in VHSC villages than in non-VHSC villages (*Sah et al., 2013*; *Srivastava et al., 2016*). Unfortunately, about 28% of all sub-centers do not have a VHSC constituted in their area. Furthermore, about 12% of all sub-centers that do have VHSCs in their area are not monitored on a regular basis. Given the evidence that regular monitoring of the ANMs' work by the VHSC is associated with an increase in the volume of health services delivered, VHSCs should be in place for all sub-centers in the country; the VHSCs should also regularly monitor the ANMs' work (*Kumar & Prakash, 2013*).

The quality of maternal health care services at sub-centers largely depends on the effectiveness with which health workers discharge their responsibilities; this, in turn, would mainly depend on the workers' training, particularly their in-service training (*Giri et al., 2012*). There are two types of in-service training programs for ANMs that are relevant to the maternal health services provided at the sub-center level. The first is known as integrated skills development training (ISDT), which is meant to upgrade the ANM's clinical, managerial, and communication skills. The second type of training is referred to as SBAT, which is intended to enable ANMs to successfully perform normal deliveries. The results of this study show that both types of training are associated with a higher volume of maternal service utilization at the sub-center. This finding is consistent with the results of two recent reports that investigated the utilization of MCH services at publicly funded health facilities in India (*Kumar & Dansereau, 2014*; *Sharma, Sharma & Livesley, 2014*). Unfortunately, the descriptive results show that the ANMs in more than 51% of sub-centers did not undergo IST even once in the last five years. The case is similar for SBAT, where the figure is even higher (67% of sub-centers). Therefore, the government should take steps to implement these trainings at regular intervals.

Another important variable that is significantly associated with the volume of maternal health services is the distance of the ANM's actual residence from the sub-center village. The effect is particularly strong in the case of delivery care. The descriptive statistics show that about 45% of ANMs live more than five kilometers away from their sub-center's village. Health workers living away from the health facility where they work is associated with increased employee absenteeism and reduced hours of work which, in turn, affect the quality and quantity of the health services provided (*Mahapatra et al., 2012*; *Muralidharan et al., 2011*). Hence, it is not surprising that the utilization of maternal health care services at those sub-centers with non-resident ANMs is lower than at those sub-centers where the ANM lives within proximity of the sub-center's village.

This study has several strengths. First, it contributes to the scant literature that has investigated the effect of facility characteristics (supply-side factors) on maternal health care utilization at public health facilities in rural India. Since most of the basic MCH services are provided through HSCs, it is imperative to understand the barriers that restrict a patient's utilization of maternal health services at this particular level of the public health system in rural areas. Second, this study adopted a large-scale nationally representative survey; hence, the results presented herein are fairly reliable and can be generalized to the national level. Third, the application of a count model–i.e., a negative binomial model–provides us with better estimates than other count models, particularly featuring variables like the ones used in this study; here, a count model was the best model to use in this context.

This study has a few limitations as well. Firstly, the analysis does not include any indicators that represent the availability of private sector health care services in the sub-center area. It is well known that a considerable amount of the population in rural areas seeks health care services in private facilities, arguably because of the poor quality of services in public health facilities. However, due to the lack of variables in the dataset that pertained to private health facilities, the analysis could not control for private health facilities in the analysis. For the same reason, the study was not able to incorporate any quality of care-related variables (e.g., health workers' behavior toward patients). Secondly, all demand-side variables were calculated at the district-level; therefore, they may not be truly representative of the population's characteristics in the sub-centers' catchment areas. Thirdly, the variable 'number of antenatal registrations' does not represent the actual number of antenatal care services provided at the health facilities–a measure that is more important from a policy perspective. Fourthly, since the data used in the study were collected using face-to-face interviews, it is also likely that the data suffer from response bias, which arose from respondents' inability to accurately answer questions and their unwillingness to respond honestly. Lastly, since the study is based on cross-sectional data, causality could not be inferred. One needs to conduct a panel data study to address the issue of reverse causality or temporality.

## CONCLUSION

Apart from highlighting the fact that a considerable number of HSCs are still understaffed and inadequately equipped, this study found that essential obstetric drugs, blood pressure instruments, labor/examination tables, bed screens, and basic amenities such as water, electricity, toilets, etc., are some important supply-side factors that affect the utilization of health services at HSCs in rural India. Other factors include the VHSC's ability to monitor the ANMs' work, the provision of in-service training for ANMs, and distance of an ANM's residence from the sub-center's village. Unlike socioeconomic factors, the supply-side factors mentioned above can be easily influenced by government interventions in a short period of time. Hence, future government interventions to improve maternal health care utilization in rural areas should consider these factors. Although this study has answered many questions, some unanswered questions also remain. For instance, why is the volume

of service utilization in those HSCs run solely by contractual ANMs more likely to be lower than in those HSCs run solely by permanent ANMs? Future studies can examine this anomaly in detail to determine why the presence of contractual ANMs is associated with lower service utilization. Since the utilization of maternal health care services is shaped by a complex web of supply and demand factors featuring a multitude of interactions, more research is needed to tease out these complex relationships.

### Funding
The author received no funding for this work.

### Competing Interests
The author declares there are no competing interests.

### Author Contributions
- Aditya Singh conceived and designed the experiments, performed the experiments, analyzed the data, contributed reagents/materials/analysis tools, wrote the paper, prepared figures and/or tables, reviewed drafts of the paper.

### Data Availability
  The raw data has been supplied as Supplemental File.

### Supplemental Information
Supplemental information for this article can be found online at http://dx.doi.org/10.7717/peerj.2675#supplemental-information.

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
