# Peer review of "Supply-side barriers to maternal health care utilization at health sub-centers in India"

_PeerJ, doi:10.7717/peerj.2675_

## Round 0.1 · original submission · Major Revisions

Please pay special attention to the reviewers' comments on the methodology of the study.

Reviewer 1 ·

Basic reporting

The author analyzes cross-sectional data from the Indian Public Health Service to determine the relationship between supply-side determinants of health and the volume of maternal health services provided. The title and abstract appropriately summarize the study while the tables and figures help guide the reader through the main findings of the study. The author uses appropriate statistical analyses to address the objectives of the study. The findings and discussion allow for further thought concerning issues surrounding the delivery of maternal health services in this setting. The references are relevant for the present study. Please find some minor suggestions for corrections marked with the Track Changes feature in the attached manuscript, which is otherwise well written and easy to follow. Please find detailed suggested comments below by section.

Experimental design

Methods
Data sources
1) Please briefly describe the type of sampling that the International Institute for Population Sciences (IIPS) used for the District Level Household and Facility Survey (DLHS-3) in this section. The author listed the link for the website of the survey but a brief mention of the sampling scheme would inform readers upfront.
Independent variables
2) On page 12, line 158 the author selected various covariates based on available literature. Is the study by Peters et al., 2008 one of the sources of these covariates? Please cite your source(s).
3) On page 12, line 173 I suggest that the author use weighing ‘scale’ as opposed to weighing ‘machine’.
Statistical analysis
4) All of the assumptions that the author states regarding the reason for using the negative binomial regression are sound, however, did you compare the Poisson model and negative binomial model in Stata with a likelihood ratio test in order to compare the two models and the assumption that the conditional means are not equal to the conditional variances? As far as I know, the log-transformed over-dispersion parameter (/lnalpha) is zero in a Poisson model. Stata calculates the alpha from the maximum likelihood estimate of the log of alpha. This alpha should always be greater than zero in a negative binomial model where the test that compares the Poisson to the negative binomial model, the likelihood ratio test of alpha, is set as zero. Then, depending on your chi-square value and degrees of freedom, the analysis may suggest that the alpha is non-zero and that the negative binomial model is more suitable for your data than the Poisson. I believe that it is insufficient for the author to state assumptions without a test that would verify your assumptions that the negative binomial model is more suitable for your data.

Validity of the findings

Results
1) Please report the results in the past tense. In some instances, throughout this section the author refers to the findings of the study in the present tense.
Discussion
1) Training as the author states is integral in the preparation of healthcare professionals to provide quality healthcare, however healthcare professionals should partake in regular assessments of competencies learned during medical education or practice. For example, in the study by Das et al. (1), they showed that physicians and health care workers in rural or urban settings in India provided medical care of varying quality to standardized patients, which could be attributed to the quality of medical education in India.
2) I would suggest that the author include another limitation: response bias. The IIPS stated that the response rate is not available for the cross-sectional study that you have used for your current study. Thus, it is necessary to inform the reader that there could be inherent differences between responders and non-responders of the District Level Household and Facility Survey (DLHS-3). Additionally, the IIPS collected the survey data through interviews. Although not explicitly stated, the data were most likely collected with face-to-face interviews, which introduces yet another type of bias.
Conclusion
1) Further study could elucidate factors that decrease the delivery of quality healthcare services; however, the author should state which type of study, i.e., a longitudinal study that would help to clarify issues relating to the volume of the utilization of services. It is not clear how the author can state that by providing more resources, such as bed screens, that the volume of maternal health services may increase. I believe that the author should state that further research is needed on how to best organize the resources of the health care system and how other factors that may increase the volume of health services at the sub-centers.
Tables and Figures
1) Please edit Tables 2 – 4, there are misplaced vertical lines where there is no data. Some cells in the tables have unaligned vertical lines in reference to the cells above them.

References
1. Das J, Holla A, Das V, Mohanan M, Tabak D, Chan B. In urban and rural India, a standardized patient study showed low levels of provider training and huge quality gaps. Health Aff (Millwood). 2012;31(12):2774-84. Epub 2012/12/06.

Annotated reviews are not available for download in order to protect the identity of reviewers who chose to remain anonymous.

Reviewer 2 ·

Basic reporting

The paper is written in good English. The introduction of the paper is brief and to the point. (for more comments see general comments' section)

Experimental design

The paper does fall within the scope of the journal. The clearly defines the research question. The study provides a good description of the gap and how this study fills that gap. Methodology used is appropriate, methods section needs some additional information though (please see general comments section).

Validity of the findings

Data set used in the study comes from a large scale survey. It is widely used by researchers and governments. The conclusions are appropriately stated, connected to the original question investigated, and are supported by the results.

Additional comments

The paper is both timely and relevant. NRHM is in its 12th year since its inception in 2004, but it seems that the poor performance (in terms of health service utilization) of publicly funded health facilities is still a concern for the government. It is indeed true that most studies in the past have focused on demand-side factors and there is dearth of studies on this issue. This paper focuses on supply-side factors that could be easily addressed at the policy and strategic management level which in turn could improve healthcare utilization at publicly funded health facilities. This paper therefore could be of great relevance to policy makers.

The purpose of the paper and the problem is clearly stated. The data and methods used for the analysis seem to be appropriate. Cross-sectional dataset used in this paper is considerably large and widely used by both the government and the researchers. Given that the dependent variables show overdispersion, the use of negative binomial regression is appropriate. The conclusions of the study reflect on the key findings. The key findings of this paper are interesting and relevant to policy makers. I find the discussion quite interesting and informative aided with relevant references.

However, I have some queries and suggestions regarding methodology, model building, and model evaluation that could make this paper more useful for readers.

I think the reader needs to know more about how the three models were put together. What estimation procedures were employed. How were the models assessed? What diagnosis did you use to assess the performance of the models? At least provide residuals vs. fitted values plot, if not scale-location and quantile plot.

How did you check heteroscedasticity of residuals? Park test? Glejser test? If there was heteroscedasticity present, what did you do to sort it out? Are the coefficients used in the final table ‘robust’?

I would like to know how did you build your models? How did you choose your variables? Did you use forward selection or backward elimination or some other method such as stepwise etc.? This information, not necessarily in great detail, should be included in methodology section to give the reader an idea how the final models to came into existence.

You have mentioned using VIF for checking multicollinearity in the methodology section. I looked for detailed VIF tables for your models but couldn’t find any. You could provide a full table for VIF and tolerance in appendix.

The section describing independent variables could be presented in a table. It would save some space which you could use to enrich your method section.

Table 2 does not provide the number of observation included in the model

I think the regression tables should present overdispersion parameter (alpha or ln(alpha)) as well. It would be beneficial for readers to know what commands in Stata you used for running your regressions – nbreg, glm, menbreg, meglm.
I did not find any mention of offset variable – what is the offset in your models? You must describe your offset/exposure variable in methods section.

---

## Round 0.2 · Minor Revisions

Please pay special attention to the language and style. I would advise that you get help from a native speaker.

Reviewer 1 ·

Basic reporting

Nice work. I am pleased with the result of this paper including the incorporation of the reviewers’ suggestions for the Methods and Discussion sections of the manuscript. The author has sufficiently addressed all of the suggested comments as described in the first round review. Based on the most recent manuscript, I recommend this manuscript for publication in PeerJ.

Experimental design

Methods
Data sources
1) The author briefly described the type of sampling that the International Institute for Population Sciences (IIPS) used for the District Level Household and Facility Survey (DLHS-3) in this section. Thank you for informing readers upfront.

Validity of the findings

Statistical analysis

The author has verified the assumptions that the negative binomial model is more suitable for the study data. Additionally, the author has provided sufficient references to substantiate the test for verification.
Discussion
The author has extended the limitations to include other types of bias associated with this study, thank you.
Conclusion
This section now states that further study could elucidate factors that decrease the delivery of quality healthcare services.
Tables and Figures
The author has edited the tables and figures sufficiently.

Reviewer 2 ·

Basic reporting

I think the article has improved considerably. However, it still needs some language-related minor corrections. For instance – in abstract – data and methods – the first line needs to be corrected – “which was collected”!!! survey was collected? No, data were collected, and the survey was conducted. The results section in abstract – third line – “with the increased utilization” or with increased utilization. Line 222-230 – should be in past tense.

Experimental design

No comments

Validity of the findings

No comments

Additional comments

Except for some isolated cases of grammatical mistakes, I think the manuscripts fulfills all criteria for publication at PeerJ. I suggest the author to go through the paper once again and ensure that there are no language and grammar related mistakes/omissions.

---

## Round 0.3 · accepted · Accept

Thank you for addressing all the reviewers' comments.